# Novel Therapeutic Effects of Pterosin B on Ang II-Induced Cardiomyocyte Hypertrophy

**DOI:** 10.3390/molecules25225279

**Published:** 2020-11-12

**Authors:** Chang Youn Lee, Han Ki Park, Bok-Sim Lee, Seongtae Jeong, Sung-Ae Hyun, Jung-Won Choi, Sang Woo Kim, Seahyoung Lee, Soyeon Lim, Ki-Chul Hwang

**Affiliations:** 1Substance Abuse Pharmacology Group, Korea Institute of Toxicology, KRICT, Daejeon 34114, Korea; changyeon.lee@kitox.re.kr (C.Y.L.); sahyun@kitox.re.kr (S.-A.H.); 2Division of Cardiovascular Surgery, Department of Thoracic and Cardiovascular Surgery, Severance Cardiovascular Hospital, Yonsei University College of Medicine, Seoul 03722, Korea; HANK@yuhs.ac; 3Institute for Bio-Medical Convergence, College of Medicine, Catholic Kwandong University, Gangneung-si, Gangwon-do 210-701, Korea; alsgur04@nate.com (B.-S.L.); 91seongtae@gmail.com (S.J.); jungwonjian@gmail.com (J.-W.C.); doctor7408@gmail.com (S.W.K.); sam1017@ish.ac.kr (S.L.)

**Keywords:** cardiomyocyte hypertrophy, Pterosin B, angiotensin II

## Abstract

Pathological cardiac hypertrophy is characterized by an abnormal increase in cardiac muscle mass in the left ventricle, resulting in cardiac dysfunction. Although various therapeutic approaches are being continuously developed for heart failure, several studies have suggested natural compounds as novel potential strategies. Considering relevant compounds, we investigated a new role for Pterosin B for which the potential life-affecting biological and therapeutic effects on cardiomyocyte hypertrophy are not fully known. Thus, we investigated whether Pterosin B can regulate cardiomyocyte hypertrophy induced by angiotensin II (Ang II) using H9c2 cells. The antihypertrophic effect of Pterosin B was evaluated, and the results showed that it reduced hypertrophy-related gene expression, cell size, and protein synthesis. In addition, upon Ang II stimulation, Pterosin B attenuated the activation and expression of major receptors, Ang II type 1 receptor and a receptor for advanced glycation end products, by inhibiting the phosphorylation of PKC-ERK-NF-κB pathway signaling molecules. In addition, Pterosin B showed the ability to reduce excessive intracellular reactive oxygen species, critical mediators for cardiac hypertrophy upon Ang II exposure, by regulating the expression levels of NAD(P)H oxidase 2/4. Our results demonstrate the protective role of Pterosin B in cardiomyocyte hypertrophy, suggesting it is a potential therapeutic candidate.

## 1. Introduction

Pathological cardiac hypertrophy is characterized by the thickening of cardiac muscle, which leads to irreversible morphological and functional changes and is mainly observed in patients with end-stage heart disease. The pathological hypertrophic heart is closely associated with the development of heart failure, arrhythmias, cerebrovascular disease, sudden death, etc [1]. Pressure overload, such as hypertension, is one of the major causes of pathological cardiac hypertrophy, in which angiotensin II (Ang II) has been reported to play a pivotal role [2,3]. Ang II, a vasoconstrictor peptide of the renin-angiotensin system (RAS), plays an important role in the regulation of blood pressure homeostasis [4]. Under pathological circumstances, Ang II has been extensively shown to be closely related to cardiac remodeling, such as myocardial hypertrophy and vascular hypertrophy, which mainly acts through Ang II type 1 receptor (AT1R) in the human and animal cardiovascular system [5,6,7]. AT1R activated by Ang II binding can activate various intracellular protein kinases, such as mitogen-activated protein kinase (MAPK) family proteins, protein kinase C (PKC), and protein kinase B (Akt) [8] and produce reactive oxygen species (ROS) via NAD(P)H oxidase (NOX) activation, which at high levels, leads to the activation of downstream signaling pathways that can stimulate cardiac hypertrophy, fibrosis, apoptosis, etc. [9]. In these Ang II-induced signaling pathways, activation of nuclear factor-κB (NF-κB) as a key transcriptional factor for inflammatory mediators is critical for the pathology of hypertension and its associated complications, such as cardiac hypertrophy [10]. Our previous data revealed that AT1R activation can stimulate receptors for advanced glycation end products (RAGE) via extracellular secretion of high-mobility group box 1 (HMGB1), in which HMGB1 binds the cytosolic domain of RAGE, leading to the activation of RAGE-mediated signaling in cardiomyocyte hypertrophy [11]. In addition to RAGE, it has been recently reported that Ang II stimulation is involved with Toll-like receptor 2 (TLR2)- and TLR4-dependent pathways in cardiac fibrosis, hypertrophy, and dysfunction [12,13,14,15]. Moreover, HMGB1 is known to bind TLR2 and TLR4, which recruit MyD88 to activate MAPK and NF-κB [16,17].

Bracken fern, *Pteridium aquilinum*, as one of the oldest and most abundant plants on Earth, has been consumed as food by many cultures, especially in East Asia [18]. To date, many reports have revealed the mainly toxic, carcinogenic, and cytotoxic effects of brackens, especially those related to ptaquiloside [19,20]. However, various beneficial effects of brackens have also been reported, including their ability to stabilize blood pressure and their immunomodulatory, anti-inflammatory, and antioxidant effects [21,22,23,24,25], which are due to brackens having various beneficial components, such as proteins, carbohydrates, fat, vitamins, carotenoids, etc. Among these beneficial components, Pterosin B, a secondary metabolite of *P. aquilinum* and known as a final product of ptaquiloside, was recently identified by its biological function; namely, Pterosin B prevents chondrocyte hypertrophy by inhibiting Sik3 [26]. However, the potential biological and therapeutic roles of Pterosin B on life-sustaining functions are not clear. Moreover, the effects of Pterosin B on heart disease have never been studied. Thus, we investigated whether Pterosin B can regulate cardiomyocyte hypertrophy induced by Ang II in vitro.

## 2. Results

### 2.1. Anti-Hypertrophic Effect of Pterosin B on Ang II-Treated Cardiomyocytes

Ang II is known as one of the major factors inducing cardiac hypertrophy [6]. Thus, we induced cardiomyocyte hypertrophy by treatment with 300 nM Ang II using H9c2 cells, an embryonic rat heart-derived cell line [11]. As Pterosin B has rarely been studied in hypertrophic cells, we wondered whether it has an effect on cardiomyocyte hypertrophy. First, we measured the viability and cytotoxicity of cell induced by Ang II with or without Pterosin B (10–100 μM) treatment. The results showed that a 300 nM Ang II treatment did not affect cell viability (Figure 1A) or cytotoxicity (Figure 1B). Pterosin B at concentrations as high as 50 μM had no effects on Ang II-treated cells, but cell viability was decreased by approximately 30% in 100 μM Pterosin B-treated cells. However, no significant differences in induced cytotoxicity were detected with concentrations as high as 100 μM. Considering these results, we treated cells with concentrations of Pterosin B up to 50 μM to investigate its anti-hypertrophic effect. We then analyzed hypertrophy-related genes, namely, ANP, BNP, and NFATc1, in Ang II-treated cells treated with or without Pterosin B (Figure 2A). The mRNA expression levels of ANP and BNP increased approximately 1.5-fold upon 300 nM Ang II stimulation, and these increases were attenuated by Pterosin B treatment in a dose-dependent manner (Figure 2B,C). The expression of NFATc1 was increased approximately 2.5-fold, and it was also decreased by Pterosin B treatment (Figure 2D). Although Pterosin B showed a tendency to decrease all three hypertrophy-related genes, 50 μM Pterosin B administered after Ang II treatment had a particularly strong effect, significantly reducing all three genes. Therefore, we used 50 μM Pterosin B for further experiments to confirm its effects on cardiomyocyte hypertrophy. As the most representative characteristic of myocardial hypertrophy, the size of myocardial cells was measured through immunofluorescence staining (Figure 2E). The results showed that the cell surface area of the cardiomyocytes was increased by approximately 250% by Ang II, and the surface area of the cells in the Pterosin B-treated group was maintained at the level of the control cells (Figure 2F). Additionally, there was no significant change in cell surface area in the group treated with Pterosin B alone. Then, we analyzed the synthesis of new intracellular proteins, which is one of the characteristics of cardiomyocyte hypertrophy (Figure 2G). As a result of visualizing the newly synthesized protein through fluorescence staining, we confirmed that the synthesis of new protein was inhibited in the Pterosin B treatment group. Considering these results, we suggest that Pterosin B can affect Ang II-induced cardiomyocyte hypertrophy.

### 2.2. Pterosin B Attenuates AT1R-Mediated HMGB1 Secretion in Ang II-Induced Hypertrophic Cardiomyocytes

It has been reported that AT1R-mediated signaling pathways are mainly involved in Ang II-induced hypertrophic effects in the heart [27,28]. In addition, we previously demonstrated the importance of the AT1R-HMGB1–RAGE axis in Ang II-induced cardiomyocyte hypertrophy [11]. Therefore, we examined the protein expression levels of AT1R and RAGE in Ang II-treated cardiomyocytes by immunoblot analysis (Figure 3A). The protein expression levels of AT1R and RAGE in the Ang II-treated cells were increased approximately 1.5-fold compared to those of the control, showing that Pterosin B significantly attenuated these receptors (Figure 3B,C). Next, we determined the protein level of intracellular HMGB1 in Ang II-treated cell lysates and the amount of secreted HMGB1 in Ang II-treated culture medium (Figure 3D). The protein levels of HMGB1 were increased in the Ang II-treated cell lysates and culture media. The expression and secretion of HMGB1 were decreased in the Pterosin B-treated cells compared with the Ang II-treated cells (Figure 3E,F). In parallel, we investigated the protein levels of TLR2 and TLR4, which are activated by HMGB1 [17], in Ang II-induced hypertrophic cardiomyocytes. The results showed that the expression levels of TLR2 and TLR4 were increased approximately 1.5-fold in the Ang II-treated cells (Figure 3F), and the levels of TLR2 and TLR4 were decreased in the Pterosin B-treated cells. In particular, both TLR2 and TLR4 showed a tendency to respond to Ang II and Pterosin B, but only the effect on TLR2 was significant (Figure 3G,H). In addition, we examined the protein expression of MyD88, which is a as a downstream adaptor protein critical for the TLR signaling pathway [29], and the results showed that MyD88 expression was also increased by Ang II treatment and was subsequently reduced by Pterosin B treatment (Figure 3F,I). These results suggested that Pterosin B may regulate the AT1R-HMGB1-RAGE axis as well as the TLR2-MyD88 pathway in Ang II-treated cardiomyocytes.

### 2.3. Pterosin B Attenuates Ang II-Mediated Signaling Pathways

Next, we investigated the effect of Pterosin B on the AT1R downstream signaling pathway in Ang II-induced cardiomyocyte hypertrophy. The results showed that Ang II increased the PKC phosphorylation by approximately 1.7-fold compared to the level of the control group. The level of phospho-PKC was decreased and maintained at the control level in Pterosin B-treated cells. In addition, the increased levels of phosphorylated ERK in the Ang II-treated cells were decreased in the Pterosin B-treated cells (Figure 4A,B). We further investigated the phosphorylation of NF-κB, a major transcription factor in myocardial hypertrophy. The phosphorylation level of NF-κB was increased approximately 1.5-fold in the Ang II-treated cardiomyocytes, whereas Pterosin B reduced the phosphorylation level of NF-κB to a level similar to that of the control group (Figure 4C,D). These results suggested that Pterosin B can modulate myocardial hypertrophy by regulating the PKC-ERK-NF-κB signaling pathway in Ang II-stimulated hypertrophic cardiomyocytes.

### 2.4. Pterosin B Attenuates Ang II-Induced Intracellular ROS

In the cardiovascular system, the AT1R-NOX axis generates cytosolic ROS upon Ang II stimulation, and these ROS activate various hypertrophic signaling factors, such as MAP kinase and NF-kB [30,31,32]. Therefore, we investigated whether Pterosin B can control cytosolic ROS production in Ang II-stimulated cardiomyocytes. The cells treated with Ang II with or without Pterosin B were incubated with the ROS indicator H_2_DCFDA, and intracellular ROS levels were assessed through confocal microscopy (Figure 5A). The cytosolic ROS level of Pterosin B-pretreated cells after Ang II stimulation for 2 h was maintained to a level similar to that of the control group, although the cytosolic ROS level was increased by approximately 15% in the Ang II-treated cardiomyocytes (Figure 5A,B). In addition, the expression levels of NOX2 and NOX4, which are the major isoforms in cardiomyocytes [33], were evaluated by immunoblotting (Figure 5C). The expression levels of NOX2 and NOX4 were increased in the Ang II-induced cardiomyocytes by approximately 35% and 30%, respectively (Figure 5D,E). The expression of NOX2 and NOX4 was significantly decreased in the Pterosin B-treated cells. Based on these results, we suggest that Pterosin B regulates intracellular ROS levels by regulating NOX2 and NOX4 expression.

## 3. Discussion

Our data showed that Pterosin B attenuates AT1R-mediated signaling pathways upon Ang II-stimulation of hypertrophic cardiomyocytes, in which Pterosin B inhibits the increase in cytosolic HMGB1 protein levels and secretion into the extracellular environment that leads to the activation of RAGE and its downstream signaling. It has been suggested that Ang II increases the protein levels of RAGE, TLR2, TLR4, HMGB1, and even AT1R itself through NF-κB activation, resulting in a positive feedback loop [34,35]. In these experiments, Pterosin B blocked the positive feedback loop by inhibiting PKC-ERK-NF-κB activation. An in vitro assay suggested that Pterosin B (50–500 μM) may not be able to bind RAGE directly (Appendix A). We also confirmed that Pterosin B suppresses Ang II-induced excessive production of ROS, which are critical mediators of signaling pathways that lead to cardiac hypertrophy (Figure 6). It has been reported that ROS production is mediated through PKC and NOX activation upon Ang II stimulation. Similar to our findings on NOX expression levels, various studies have supported the association of NOX activity and expression levels with ROS production. Dai et al. emphasized the roles of NOX2 and NOX4 to produce intracellular ROS by Ang II-stimulated AT1R, depending on their intracellular location [36,37]. Zhang et al. demonstrated that NOX 4 expression levels also affect ROS production through the use of silencing and overexpression experiments [38], and using an overexpression in vivo model, Dai et al. also evaluated the regulatory effects of NOX2 expression levels on vascular remodeling and hypertension [39]. However, additional validation such as NADH Oxidase activity is needed to elucidate the ability of Pterosin B against intracellular ROS. In addition, a recent study revealed a new role for ROS, showing that, under hypoxic conditions and ultraviolet radiation, they can induce the secretion of HMGB1 from the nucleus into the extracellular area [40,41].

Considering the findings from our study, we suggest that extracellular HMGB1, as a downstream signal of AT1R stimulation, is one of the major mediators of RAGE activation. A ubiquitous nuclear protein, HMGB1 acts as a damage-associated molecular pattern (DAMP) upon passive or active secretion from damaged or stressed cells [42]. Various studies have shown the critical roles of extracellular HMGB1 in cardiovascular diseases, including myocarditis and cardiomyopathies [43,44]. Our previous studies also showed that an anti-HMGB1 antibody attenuated RAGE expression and its downstream effector NF-κB activation in Ang II-induced endothelial dysfunction and cardiomyocyte hypertrophy [11,34]. Although the mechanism of extracellular HMGB1 secretion upon Ang II-induced cardiomyocyte hypertrophy is poorly understood, Oh et al. demonstrated that classical PKC, not ERK or NF-κB, can phosphorylate HMGB1, resulting in extracellular HMGB1 secretion from LPS-stimulated Raw264.7 cells, and An et al. showed PKC-induced phosphorylation and secretion of HMGB1 using a bone cancer pain model [45,46]. Regarding HMGB1 secretion in an Ang II-induced environment, Zhou et al. recently provided another novel mechanism, in which Ang II can induce HMGB1 release by increasing the hyperacetylation of HMGB1, by separation from a complex of HMGB1 and SIRT1 in Raw264.7 cells [47]. To address the possible mechanisms controlling HMGB1 release in Ang II-induced cardiomyocyte hypertrophy, we need further experiments using a PKC inhibitor or an anti-SIRT1 antibody.

Finally, we further assessed hERG assay to predict the possible use of Pterosin B as a pharmaceutical compound. Pterosin B at 50 μM showed the hERG inhibition by approximately 6.7% (Appendix A). As a cardiac repolarizing potassium channel, the hERG channel has been reported to be associated with long QT syndrome due to unintended blocking [48]. As diverse chemicals can bind and block the hERG channel, several agents, such as terfenadine and astemizole, have been withdrawn from the market because of the potential occurrence of sudden cardiac death [49,50]. Therefore, further study using the new synthesis of Pterosin B derivatives is necessary to overcome this safety issue.

## 4. Materials and Methods

### 4.1. Reagent

The Pterosin B was purchased from Santa Cruz Biotechnology (catalog number: sc-476730, Dallas, TX, USA) and dissolved in DMSO before use.

### 4.2. Induction of Cardiomyocyte Hypertrophy In Vitro

H9c2 cells (Korean Cell Line Bank, Seoul, Korea) were cultured in Dulbecco’s modified Eagle’s medium (DMEM; Gibco; Thermo Fisher Scientific, Inc., Waltham, MA, USA) supplied with 10% Fetal Bovine Serum (FBS; Atlas Biologicals, Fort Collins, CO, USA) and 1% Penicillin-Streptomycin (P/S; Gibo, Thermo Fisher Scientific, Waltham, MA, USA) at 37 °C under humidified conditions (5% CO_2_ atmosphere). The cells were cultured for 24 h in DMEM supplied with 0.5% FBS to expose the cells to starvation. To induce hypertrophy, 300 nM Ang II is treated in the cells for 48 h then, the hypertrophy was examined.

### 4.3. Cell Viability/Cytotoxicity Assay

For analyze the cell viability / cytotoxicity, 5 × 10^3^ cells per well were seeded in 96-well plates and incubated for 24 h at 37 °C under humidified conditions (5% CO_2_ atmosphere). Then, cells were incubated for 24 h in a DMEM containing 0.5% FBS. The cells were treated with Pterosin B at concentrations of 10, 25, and 50 μM for 1 h, and Ang II was then treated for 48 h.

#### 4.3.1. Cell Viability

Then, EZ-Cytox Kit (DoGen, Seoul, Korea) is an assay kit based on water soluble tetrazolium salt (WST) assay. The assay reagent was added to each well at final concentration of 0.5 mg/mL and the cells were incubated for 2 h at 37 °C under humidified conditions (5% CO_2_ atmosphere). To check cell viability, the absorbance was measured at 450 nm by a microplate reader (Thermo Fisher Scientific, Waltham, MA, USA).

#### 4.3.2. Cytotoxicity Assay

The experiment was conducted using Lactate dehydrogenase (LDH) Cytotoxicity Detection Kit (Takara Bio Inc, Kusatsu, Japan), according to a manufacturer’s protocol. Briefly, the Pterosin B-treated cells were incubated for 48 h then and 100 μL of cell-culture supernatant was transferred into optically clear 96-well flat bottom microtiter plate. To determine the LDH activity, 100 μL of reaction mixture was added and incubated for 30 min at room temperature. The absorbance at 490 nm was measured using a microplate reader (Thermo Fisher Scientific, Waltham, MA, USA). The absorbance value of assay medium as background control was subtracted from other values of test samples.

### 4.4. Immunocytochemistry and Cardiomyocyte Cell Size Measurement

2 × 10^4^ cells per well were seeded in 4-well chamber slide and induced cardiomyocyte hypertrophy by Ang II treatment. 48 h after the incubation, the cells were fixed with 4% paraformaldehyde for 10 min at room temperature, it was then washed twice with PBS and was permeabilized with 0.2% Triton X-100 for 10 min. Next, it was washed with PBS, was blocked with blocking solution (1% bovine serum albumin) for 1 h, and then was stained with Texas Red™-X Phalloidin (Thermo Fisher Scientific, Waltham, MA, USA). Immunofluorescence was detected by confocal microscopy (LSM 700; Carl Zeiss, Wetzlar, Germany). More than 50 cells in 10 randomly selected regions per experiment were measured in six independent experiments. The cell size was measured using ImageJ software, version 1.48v (National Institutes of Health, Bethesda, MD, USA).

### 4.5. Reverse Transcription PCR Analysis

The level of each gene transcript was quantified using polymerase chain reaction (PCR). Total RNA was extracted from a 60 mm culture dish using Hybrid-R (Total RNA isolation kit; GeneAll, Seoul, Korea) according to the manufacturer’s instructions then, cDNA synthesis was performed using a Maxime™ RT Premix kit (iNtRON Biotechnology, Seongnam, Korea). The EmeraldAmp^®^ GT PCR Master Mix (Takara Bio Inc, Kusatsu, Japan) was used to perform PCR. The level of each gene transcript was normalized to GAPDH. PCR primers are listed as follows: ANP forward primer: 5′- TGA GCC GAG ACA GCA AAC-3′; ANP reverse primer: 5′-CTC ATC TTC TAC CGG CAT CTT C-3′; BNP forward primer: 5′-TCT GCT CCT GCT TTT CCT TA-3′; BNP reverse primer: 5′-GAA CTA TGT GCC ATC TTG GA -3′; NFATC1 forward primer: 5′-TCC ACG ATG TGG AGG TGG AAG ACG-3′; NFATC1 reverse primer: 5′-TGA TGG CTG CCA CAA TAG CAG AGC-3′; GAPDH forward primer: 5′-GGT CTT GCT GTG GAC TGG AT-3′; GAPDH reverse primer: 5′-CTG CTA CAG CAA ATG GGT GA-3′.

### 4.6. Quantification of Protein Synthesis

To measure protein synthesis, we used Click-iT™ HPG Alexa Fluor™ 594 Protein Synthesis Assay Kit (Thermo Fisher, Waltham, MA, USA). Cells were incubated in L-methionine-free DMEM (Gibco, Thermo Fisher Scientific, Waltham, MA, USA) for 30 min prior to the addition of 50 μM methionine analog L-homopropargylglycine (HPG) for 30 min. After fixation and permeabilization, the Click-iT reaction using Alexa-488 was performed according to the manufacture’s protocol. Immunofluorescence was detected via confocal microscopy (LSM700; Carl Zeiss, Jena, Germany) Then, immune-fluorescence intensities were measured and analyzed using Zen software (Carl Zeiss, Jena, Germany), in which around 5–10 randomly selected regions per experiment were measured in three independent experiments and fluorescence intensities of each group were normalized by number of 4′,6-diamidino-2-phenylindole (DAPI)-positive cells.

### 4.7. Immunoblot Analysis

Cells were washed once with PBS and lysed in RIPA buffer (Thermo Scientific Inc. Waltham, MA, USA). Protein concentration was measured using a bovine serum albumin standard (Thermo Scientific Inc. Waltham, MA, USA). The proteins were then separated in a 10% sodium dodecyl sulfate-polyacrylamide gel and transferred to a polyvinylidene difluoride membrane (EMD Millipore, Burlington, MA, USA). After blocking the membrane using 10% nonfat dried milk diluted with Tris-buffered saline-Tween 20 (TBS-T, 0.1%) for 1 h at room temperature, the membrane was shortly washed with TBS-T twice and incubated with the primary antibody overnight at 4 °C. The following antibodies were used in these experiments: anti-MEK1/2 (#9122, Cell Signaling, Danvers, MA, USA), anti-phospho-MEK1/2 (#9121, Cell Signaling), anti-ERK1/2 (#9102, Cell Signaling), anti-phospho-ERK1/2 (#9101, Cell Signaling), anti-phospho-PKC (#9371, Cell Signaling), anti-PKC (#610127, Cell Signaling), anti-phoshpo-NFkB (#6956S, Cell Signaling), anti-NFkB (#3033S, Cell Signaling), anti-AT1R (#AAR-011, Alommone Labs, Jerusalem, Israel), anti-RAGE (#ab225533, Abcam, Cambridge, UK), anti-TLR2 (#ab213676, Abcam, Cambridge, UK), anti-TLR4 (#NB100-56566, Novus, Centennial, CO, USA), anti-Myod88 (#LS-C384717-100, LSBio, Seattle, WA, USA), anti-NOX2 (#ab129068, Abcam, Cambridge, UK), anti-NOX4 (#ab109225, Abcam, Cambridge, UK), and anti-β-actin (A5316, Sigma Aldrich, St. Louis, MO, USA). The membrane was rinsed three times with 0.1% TBS-T for 5 min then, was incubated with peroxidase-conjugated secondary antibodies (Santa Cruz Biotechnology, Dallas, TX, USA) for 1 h at RT. After the membrane was rinsed six times for 5 min each, the bands were detected with the AmershamECL Prime Western Blotting Detection Reagent (GE Healthcare, Little Chalfont, UK). The band intensities were obtained by a Davinch-Western imaging system (Davinch K, Seoul, Korea) and were analyzed using ImageJ software, version 1.48v (National Institutes of Health, Bethesda, MD, USA).

### 4.8. Measuring Cytosolic ROS Levels

Cytosolic ROS level was measured in 300 nM Ang II with or without treatment of 50 μM Pterosin B treated cells. The cells were plated in 24-well plates. Next day, 50 μM Pterosin B was treated for 1 h followed by 2 h treatment of 300 nM Ang II. Next, the cells were washed with PBS twice, and were resuspended in pre-warmed PBS containing 10 μM CM-H2DCFDA dye (C6827, Thermo Fisher Scientific, Waltham, MA, USA) for ROS detection. The cells were then incubated at 37 °C for 10 min to render the dye responsive to oxidation. Fluorescence images were obtained from 6 independent experiments (10–15 randomly selected regions per each experiment). Fluorescence intensities were measured using CELL IMAGING CELENA^®^ X high content imaging system (CX 30000, Logos Biosystems, Anyang, Korea).

### 4.9. Statistical Analysis

The data was expressed as the mean ± SEM. The significance of differences between two groups was compared using Student’s t-test. Comparisons of more than two groups were performed via one-way ANOVA with Bonferroni correction. *P* values were indicated in figure legends.

## 5. Conclusions

In the present study, we investigated the effects of Pterosin B on Ang II-induced cardiomyocyte hypertrophy, in which Pterosin B attenuated activation of major hypertrophic signaling pathways and also can regulate excessive cytosolic ROS production in the presence of Ang II. However, additional in depth study to elucidate the underlying mechanisms in details and to validate its therapeutic efficacy using in vitro and in vivo models.

## Figures and Tables

**Figure 1 molecules-25-05279-f001:**
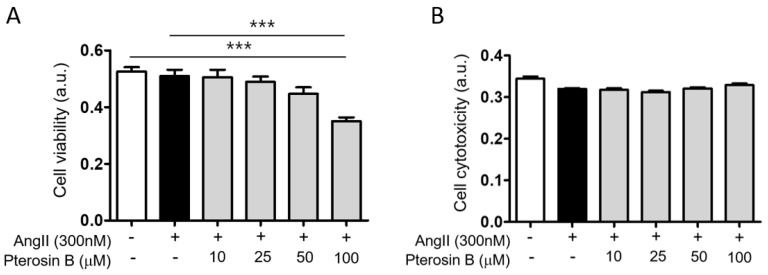
Effects of Pterosin B on H9c2 cell viability. (**A**) Cell viability was measured by WST assay, and (**B**) cytotoxicity was measured by LDH assay. Cell viability (CCK) and cytotoxicity (LDH) were measured after treatment with 0–100 µM Pterosin B for 48 h in the presence of Ang II. Quantitative data were expressed as the mean ± SEM of three independent experiments. *** *p* < 0.001.

**Figure 2 molecules-25-05279-f002:**
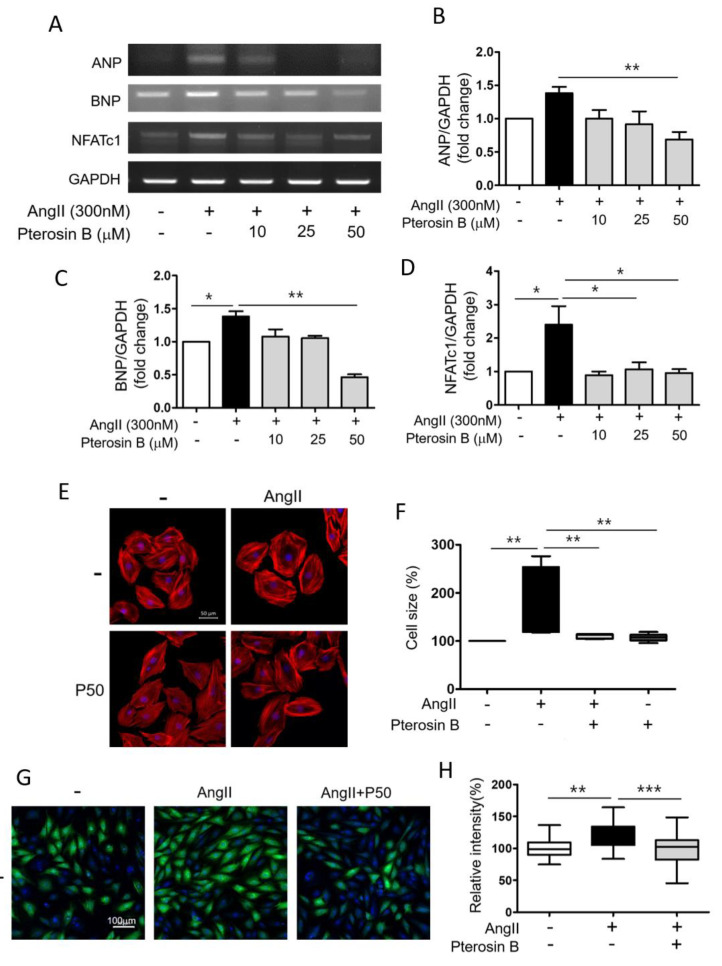
Anti-hypertrophic Effect of Pterosin B on Ang II-stimulated H9c2 cells. H9c2 cells were treated with 10–50 µM Pterosin B for 48 h in the presence of 300 nM ANG II. (**A**) Expression levels of cardiomyocyte hypertrophy-related mRNAs (ANP, BNP, and NFATc1) were estimated by reverse transcriptase PCR. (**B**–**D**) The ANP, BNP, and NFATc1 mRNA expression levels were normalized by GAPDH. * *p* < 0.05, ** *p* < 0.01. *n* = 3. (**E**) Texas-Red X-tagged phalloidin was stained for F-actin in H9c2 cells. Representative image was detected by confocal microscopy. P50 = Pterosin B (50 μM). Scale bar = 50 µm. (**F**) Average cell size was measured in each group and analyzed by Image J software. ** *p* < 0.01. *n* = 6. (**G**) Protein synthesis of H9c2 cells was determined using Click-iT^®^ HPG Alexa Fluor^®^ 488 Protein Synthesis Assay Kit. Green fluorescence indicates synthesis of new intracellular proteins and blue fluorescence (DAPI) indicates nucleus. P50 = Pterosin B (50 μM). Scale bar = 100 µm. (**H**) Quantification of the fluorescence intensity of protein synthesis. ** *p* < 0.01, *** *p* < 0.001. Scale bar = 100 µm. *n* = 4.

**Figure 3 molecules-25-05279-f003:**
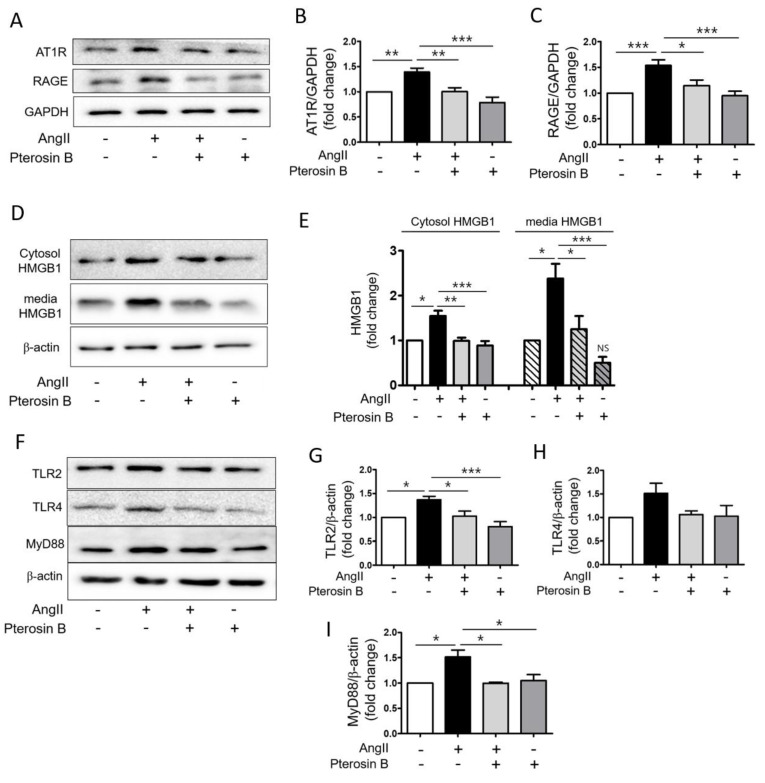
Effects of Pterosin B on Ang II-activated HMGB1 secretion. H9c2 cells were treated with 300 nM Ang II with or without Pterosin B. (**A**) Protein expression levels of AT1R and RAGE in H9c2 cells-treated with 300 nM Ang II with or without Pterosin B for 48 h. (**B**,**C**) Quantification of AT1R and RAGE expression levels shown in (**A**). The protein levels were normalized to GAPDH. * *p* < 0.05, ** *p* < 0.01, *** *p* < 0.001. *n* = 6. (**D**) H9c2 cells were treated with Ang II for 10 min and HMGB1 expression level was detected by Western blot from cell lysate and culture medium. (**E**) Quantification of intracellular HMGB1 protein levels and amount of secreted HMGB1 shown in (**D**). * *p* < 0.05, ** *p* < 0.01, *** *p* < 0.001. NS vs. control. *n* = 4. (**F**) H9c2 cells were treated with Ang II with or without Pterosin B for 48 h. Western blot was performed to detect protein expression levels of TLR2, TLR4, and MyD88. (**G**–**I**) Quantification of protein levels shown in (**F**). * *p* < 0.05, *** *p* < 0.001. TLR2 (*n* = 6), TLR4 (*n* = 6), and MyD88 (*n* = 3).

**Figure 4 molecules-25-05279-f004:**
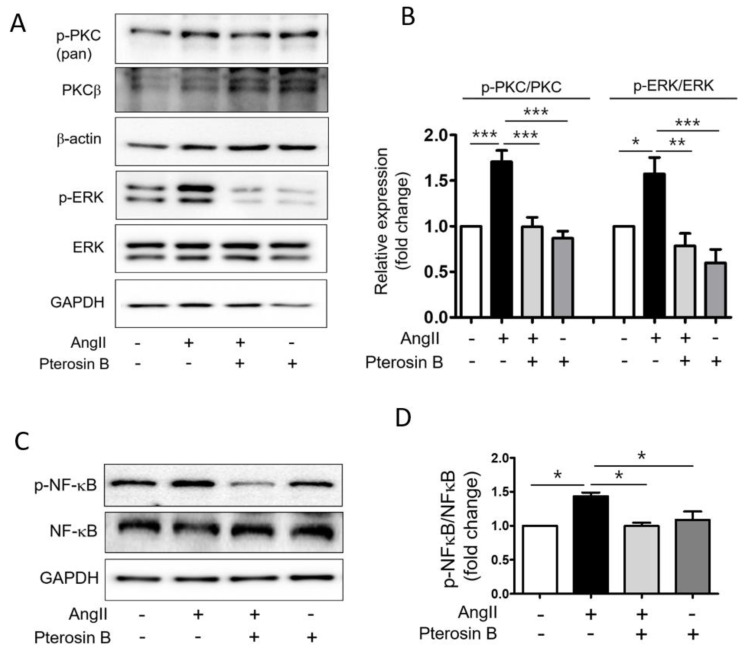
Pterosin B attenuates PKC-ERK activation in Ang II-induced cardiomyocyte hypertrophy. (**A**) H9c2 cells were treated with 300 nM Ang II for 10 min with or without Pterosin B and protein levels were analyzed by western blotting. (**B**) The values of the relative p-PKC (*n* = 6) and p-ERK (*n* = 7) were normalized by PKC and ERK, respectively. * *p* < 0.05, ** *p* < 0.01, *** *p* < 0.001. (**C**) H9c2 cells were treated with 300 nM Ang II for 2 h in the presence and absence of Pterosin B, and protein levels of NF-κB and p-NF-κB were examined by western blotting. (**D**) Quantification of protein levels shown in (**C**). * *p* < 0.05. *n* = 3.

**Figure 5 molecules-25-05279-f005:**
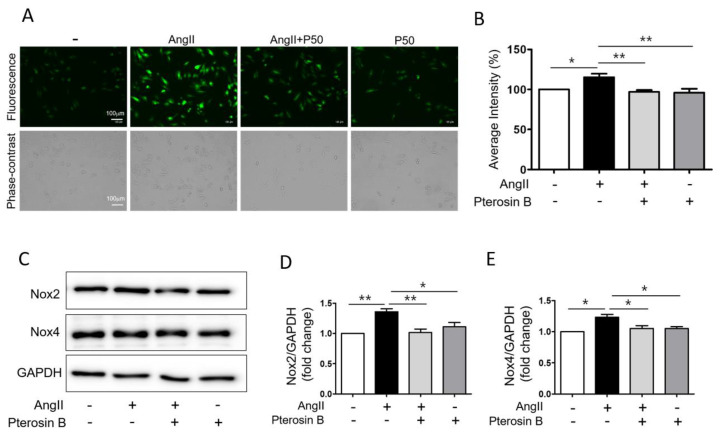
Anti-oxidant effect of Pterosin B in Ang II-treated H9c2 cells. (**A**) H9c2 cells were treated with 300 nM Ang II for 2 h with or without Pterosin B. The intracellular ROS levels were detected by CM-H_2_DCFDA (upper panel) and phase-contrast image was obtained to confirm cell morphology (lower panel). P50 = Pterosin B (50 μM). Scale bar = 100 µm. (**B**) Quantification of the fluorescence intensity of H9c2 cells in each condition. *n* = 6. (**C**) H9c2 cells were treated with 300 nM Ang II with or without Pterosin B for 2 h and the protein expression of NOX2, NOX4, and GAPDH was examined by western blotting. (**D**–**E**) Quantification of protein levels shown in (**C**). The levels of protein were normalized to GAPDH expression. * *p* < 0.05, ** *p* < 0.01. *n* = 3.

**Figure 6 molecules-25-05279-f006:**
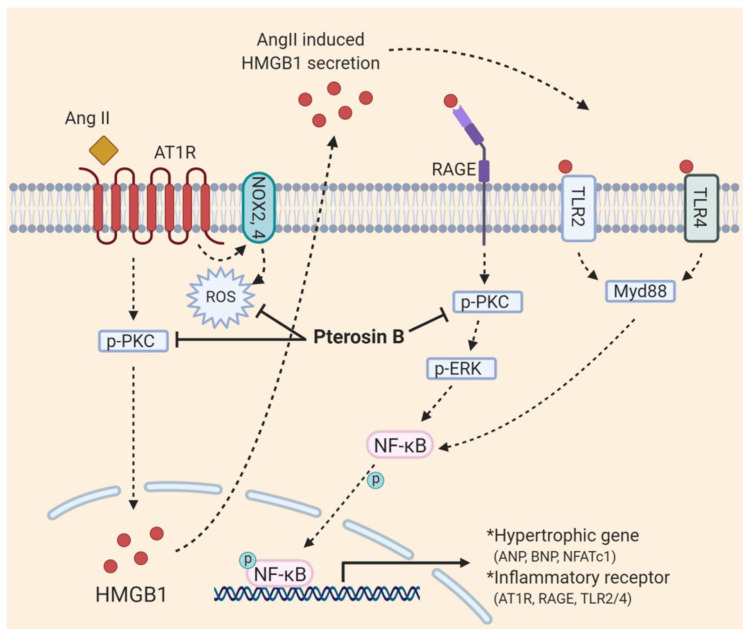
Schematic diagram with roles of Pterosin B in Ang II-induced cardiomyocyte hypertrophy.

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
