# Peer review of "Novel Therapeutic Effects of Pterosin B on Ang II-Induced Cardiomyocyte Hypertrophy"

_molecules, 2020, doi:10.3390/molecules25225279_

Round 1
Reviewer 1 Report
Lee et al investigated the role of Pterosin B on cardiomyocyte hypertrophy induced by angiotensin II, using H9c2 cell line.
The data presented in this work are of potential interest, but before considering the possibility of publishing the study in Cancers, significant changes in manuscript should be made. Overall, authors cannot conclude based on their experiments with just one cell line and without mechanistic approaches.
1) Figure 1. “Effects of Pterosin B on H9c2 cell viability. (A) Cell viability was measured by CCK-8 assay, and (B) cytotoxicity was measured by LDH assay. Cell viability (CCK) and cytotoxicity (LDH) were measured after treatment with 0-100 μM Pterosin B for 48 h in the presence of Ang II”.
Does it mean that the action of Pterosin B did not extends for longer than 48 hours?
2) 4.3.1. Cell viability
“Then, EZ-Cytox Kit (DoGen, Seoul, Korea) is an assay kit based on water soluble tetrazolium 286 salt (WST) assay”. Does this assay corresponded to CCK-8 assay, as described in Figure 1?
3) 4.3.2. Cytotoxicity assay
“The experiment was conducted using Lactate dehydrogenase (LDH) Cytotoxicity Detection Kit (Takara Bio Inc., Japan), according to a manufacturer's protocol. Briefly, the Pterosin B-treated cells 293 were incubated for 48 h then, the absorbance at 490 nm was measured using a microplate reader (Thermo Fisher Scientific). The authors need to better explain this experimental procedure, to allow it to be repeated.
4) Measuring cytosolic ROS levels
Authors declared that “Pterosin B attenuated activation of major hypertrophic signaling pathways and excessive ROS production in the presence of Ang II”. Related to its antioxidant activity. The quantification of oxidants by CM-H2DCFDA rely on a methodology that is not entirely reliable and concerns have been raised
5) Authors should provide the full-length blots in supplementary materials.
6) Photographs are not of sufficent information content and of low resolution. The cell photos (100 µm) are hard to see, so please replace them with new ones.
7) Therefore, the conclusions of the study are a little over-reaching based on these results. Mechanistic studies are recommended to assess the role of Pterosin B on Ang II-induced cardiomyocyte hypertrophy
8) Proofreading is necessary
Author Response
29, October, 2020
Dear Editor,
We authors very much appreciated the encouraging, critical and constructive comments and suggestions on this manuscript by the reviewer. The comments have been very thorough and useful in improving the manuscript. Major and minor revised parts are highlighted in yellow color (Reviewer 1) and green color (Reviewer 2) for your convenience of reviewing.
Reviewer 1
Comments and Suggestions for Authors
Lee et al investigated the role of Pterosin B on cardiomyocyte hypertrophy induced by angiotensin II, using H9c2 cell line.
The data presented in this work are of potential interest, but before considering the possibility of publishing the study in Cancers, significant changes in manuscript should be made. Overall, authors cannot conclude based on their experiments with just one cell line and without mechanistic approaches.
à We authors first would like to make sure that we submitted in Molecules, not Cancers. We authors made changes point by point (yellow color) as reviewer’s valuable comment.
1) Figure 1. “Effects of Pterosin B on H9c2 cell viability. (A) Cell viability was measured by CCK-8 assay, and (B) cytotoxicity was measured by LDH assay. Cell viability (CCK) and cytotoxicity (LDH) were measured after treatment with 0-100 μM Pterosin B for 48 h in the presence of Ang II”. Does it mean that the action of Pterosin B did not extends for longer than 48 hours?
-Response: We treated Ang II for 48 h to induce cardiomyocyte hypertrophy, which was the longest incubation time. The treatment time of Pterosin B was like that. Therefore we tested CCK-8 and LDH 48 h after Pterosin B treatment.
2) 4.3.1. Cell viability
“Then, EZ-Cytox Kit (DoGen , Seoul, Korea) is an assay kit based on water soluble tetrazolium 286 salt (WST) assay”. Does this assay corresponded to CCK-8 assay, as described in Figure 1?
-Response: Thank you for knowing us. In this article, we used EZ-Cytox kit, not a CCK-8 assay kit, although the principle of both (CCK-8 and EZ-Cytox kit) is similar. Therefore we changed the word in Figure 1 legend.
3) 4.3.2. Cytotoxicity assay
“The experiment was conducted using Lactate dehydrogenase (LDH) Cytotoxicity Detection Kit (Takara Bio Inc., Japan), according to a manufacturer's protocol. Briefly, the Pterosin B-treated cells 293 were incubated for 48 h then, the absorbance at 490 nm was measured using a microplate reader (Thermo Fisher Scientific). The authors need to better explain this experimental procedure, to allow it to be repeated.
-Response: We wrote more detail on experimental procedure (4.3.2. cytotoxicity assay).
4) Measuring cytosolic ROS levels
Authors declared that “Pterosin B attenuated activation of major hypertrophic signaling pathways and excessive ROS production in the presence of Ang II”. Related to its antioxidant activity. The quantification of oxidants by CM-H2DCFDA rely on a methodology that is not entirely reliable and concerns have been raised
-Response: As reviewer mentioned, the quantification of oxidants by CM-H2DCFDA may not enough to evaluate the anti-oxidant activity of Pterosin B. Therefore, we authors demonstrated the expression levels of NOXs isoforms (major isoforms in cardiomyocytes) which have been reported the association with ROS production. However, we also agree that other experimental estimation using FACS or NADH Oxidase activity assay kit in the near future as your suggestion (first paragraph in Discussion section and Conclusion).
5) Authors should provide the full-length blots in supplementary materials.
-Response: We added raw western blot data in supplementary materials.
6) Photographs are not of sufficient information content and of low resolution. The cell photos (100 µm) are hard to see, so please replace them with new ones.
-Response: Figure 2G and 5A represent protein synthesis and cytosolic ROS production, respectively. Therefore, we thought that overall tendency is more meaningful, and then we took pictures with low magnification. However, we added the DAPI pictures having more intense signal to more easily recognize the cells in Figure 2G and added phase-contrast pictures to correlate with cell location in Figure 5A.
7) Therefore, the conclusions of the study are a little over-reaching based on these results. Mechanistic studies are recommended to assess the role of Pterosin B on Ang II-induced cardiomyocyte hypertrophy
-Response: We authors also agree with reviewer’s recommendation. Therefore, we added the several limitations for these experiments in Discussion section concerning especially ROS estimation.
8) Proofreading is necessary
-Response: Although we got the correction from the AJE, we did carefully review our paper again as reviewer’s request (verification code 0556-92A6-5C4E-59B7-042E in AjE website).
Reviewer 2 Report
This manuscript presents an interesting study into the pharmacological effect of Pterosin B on Ang II-induced hypertrophy of cardiomyocytes. The data is presented a logical manor to follow the potential signalling pathways and the authors demonstrate that Pterosin B acts through the PKC-ERK-NFkB pathway. They conclude by showing that the compound may also have some toxic effect via hERG inhibition, and therefore should be further investigated before being considered as a pharmaceutical.
My only major concern is with the way the data are presented. Most of the graphs show the data as fold change or % change relative to untreated cells, which is fine except that there are no errors shown on the bar for the untreated cells. Even though data are normalised to this data set, there should still be variability within it. The data may be very tight but then the bar should still be shown, as for the cells treated with AngII and Pterosin B in Fig 3I. If all control cells values were set to 1 or to 100% before the statistical analysis was done, then this would give incorrect p values as the variability in the control data set would not have been included in the analysis. I hope this was not the case. The authors should confirm to the Editor that the statistical analysis was done appropriately and present the data showing the variability on the control data set, either using an error bar and/or by showing the data points.
Minor points
Section 2.2 line 152 should read Figure 3E not 3F.
In figure 3E, is there a significaint decrease in media HMGB1 in cells treated with Pterosin B alone compared with control?
In figure 5C, the expression of Nox2 in cells treated with Pterosin B alone appears lower than controls in the representative blot but not in the quantification.
In the methods, it is not clear whether the LDH cytotoxicity assay was done on media or on the cells themselves.
Author Response
29, October, 2020
Dear Editor,
We authors very much appreciated the encouraging, critical and constructive comments and suggestions on this manuscript by the reviewer. The comments have been very thorough and useful in improving the manuscript. Major and minor revised parts are highlighted in yellow color (Reviewer 1) and green color (Reviewer 2) for your convenience of reviewing.
Reviewer 2
Comments and Suggestions for Authors
This manuscript presents an interesting study into the pharmacological effect of Pterosin B on Ang II-induced hypertrophy of cardiomyocytes. The data is presented a logical manor to follow the potential signalling pathways and the authors demonstrate that Pterosin B acts through the PKC-ERK-NFkB pathway. They conclude by showing that the compound may also have some toxic effect via hERG inhibition, and therefore should be further investigated before being considered as a pharmaceutical.
My only major concern is with the way the data are presented. Most of the graphs show the data as fold change or % change relative to untreated cells, which is fine except that there are no errors shown on the bar for the untreated cells. Even though data are normalised to this data set, there should still be variability within it. The data may be very tight but then the bar should still be shown, as for the cells treated with AngII and Pterosin B in Fig 3I. If all control cells values were set to 1 or to 100% before the statistical analysis was done, then this would give incorrect p values as the variability in the control data set would not have been included in the analysis. I hope this was not the case. The authors should confirm to the Editor that the statistical analysis was done appropriately and present the data showing the variability on the control data set, either using an error bar and/or by showing the data points.
-Response: Our experimental results were obtained from more than 3 of independently performed experimental set. I double-checked our raw data of western blot and RT-PCR. For example, if we run a western blot simultaneously, raw values of control band estimated from Image J were similar at least. However, if we run the western blot using different sample set obtained different day (this means that our data was obtained independently), the basal control values from Image J have some variations in most of cases. Here are the example of Fig 3I. That’s the reason why we made control as 1.
|
Sample obtained |
|
myd88 (raw value) |
Actin(raw value) |
myod88/actin |
Conversion to ratio |
|
Apr. 17th |
con |
16443.9 |
27588.6 |
0.59604 |
1 |
|
|
angii |
21400.6 |
28000.5 |
0.76429 |
1.28229 |
|
|
ap |
17116.9 |
29435.7 |
0.5815 |
0.97561 |
|
|
p |
17800.8 |
27163.6 |
0.65532 |
1.09946 |
|
|
|
|
|
|
|
|
Apr. 28th |
con |
8420.79 |
9594.31 |
0.87769 |
1 |
|
|
angii |
14928.9 |
11346.8 |
1.31569 |
1.49905 |
|
|
ap |
11751 |
13687.7 |
0.85851 |
0.97815 |
|
|
p |
12684.2 |
11808.1 |
1.07419 |
1.22389 |
|
|
|
|
|
|
|
|
Apr. 29th |
con |
9486.28 |
12123.9 |
0.78244 |
1 |
|
|
angii |
15440.3 |
11202.8 |
1.37825 |
1.76147 |
|
|
ap |
11885.1 |
14643.4 |
0.81163 |
1.03731 |
|
|
p |
8279.4 |
12807.3 |
0.64646 |
0.82621 |
Minor points
Section 2.2 line 152 should read Figure 3E not 3F.
-Response: We corrected in section 2.2. line 152.
In figure 3E, is there a significaint decrease in media HMGB1 in cells treated with Pterosin B alone compared with control?
-Response: We looked at our raw data and statistics, but there is no significance. Figure 3E and figure legend were modified.
In figure 5C, the expression of Nox2 in cells treated with Pterosin B alone appears lower than controls in the representative blot but not in the quantification.
-Response: We changed the blot which can represent for the quantification.
In the methods, it is not clear whether the LDH cytotoxicity assay was done on media or on the cells themselves.
-Response: We clearly described more detailed method of LDH cytotoxicity assay in 4.3.2.
Round 2
Reviewer 1 Report
Authors have been submitted in Molecules, not Cancers. Sorry my misunderstanding.
However, In the Supplemntary raw data file (PDF) for WB and RT-PCR authors provided the cropped gels (as well as in the figures). The full-length blots must be provided, including molecular wheight marker. The western blot results are not convincing.
Regard to Response 4, “As reviewer mentioned, the quantification of oxidants by CM-H2DCFDA may not enough to evaluate the anti-oxidant activity of Pterosin B. Therefore, we authors demonstrated the expression levels of NOXs isoforms (major isoforms in cardiomyocytes) which have been reported the association with ROS production. However, the western blot result is not convincing, without the full-length blot.
So, authors the authors should have provided in this paper other experimental estimation using FACS or NADH Oxidase activity assay kit.
Author Response
Dear reviewer,
All changes in the article are highlighted in yellow.
Thank you very much for your valuable review.
Reviewer 1: Comments and Suggestions for Authors
Authors have been submitted in Molecules, not Cancers. Sorry my misunderstanding.
However, In the Supplementary raw data file (PDF) for WB and RT-PCR authors provided the cropped gels (as well as in the figures). The full-length blots must be provided, including molecular weight marker. The western blot results are not convincing.
Response: As we described in Materials and Methods 4.7., western blot detection was performed by Davinch-Western system. So, although we loaded and ran the protein samples and the protein size marker for WB at the same time, of course in a same SDS gel, detection time is different in most of cases. Therefore, the intensity difference between samples and protein size marker was made.
And we feel sorry to do not have whole blot for some of WB data. As we did a pretest for those Ab before making the data for the article and we wanted to detect many WB results from the same samples, some of WB blot sheet were cut many pieces.
So, I really wish that reviewer generously understand on this situation.
We authors put the whole WB blots if we had and protein size markers as Supplementary Figure S3.
Regard to Response 4, “As reviewer mentioned, the quantification of oxidants by CM-H2DCFDA may not enough to evaluate the anti-oxidant activity of Pterosin B. Therefore, we authors demonstrated the expression levels of NOXs isoforms (major isoforms in cardiomyocytes) which have been reported the association with ROS production. However, the western blot result is not convincing, without the full-length blot.
So, authors the authors should have provided in this paper other experimental estimation using FACS or NADH Oxidase activity assay kit.
Response: We authors agree your valuable suggestion that experimental data using FACS or NADH Oxidase activity assay can better support our data. However, we do not have those kit at this moment, it will be more than one month from now if we order. So, I really wish that reviewer generously understand on this situation. We authors put the whole WB blots if we had and protein size markers as Supplementary Figure S3. And to confirm NOXs WB size marker, we ran the WB again including size marker and using whole blot.